# Vaccinated COVID-19 Index Cases Are Less Likely to Transmit SARS-CoV-2 to Their Household Contacts: A Cohort Study

**DOI:** 10.3390/vaccines12030240

**Published:** 2024-02-26

**Authors:** Pere Godoy, Iván Martínez-Baz, Ignasi Parron, Manuel García-Cenoz, Joaquim Ferras, Mònica Carol, Nuria Bes, Montserrat Guillaumes, Sofia Godoy, Diana Toledo, Núria Follia, Carme Miret, Jessica Pardos, Miquel Alsedà, Pedro Plans-Rubio, Inma Sanz, Maria-Rosa Sala, Joan A. Caylà, Jacobo Mendioroz, Carmen Muñoz-Almagro, Jesús Castilla, Ángela Domínguez

**Affiliations:** 1Institut de Recerca Biomédica (IRB Lleida), Universitat de Lleida, 25006 Lleida, Spain; sofiagodoygarcia@gmail.com (S.G.); mcmiret.lleida.ics@gencat.cat (C.M.); miquel.alseda@gencat.cat (M.A.); 2CIBER de Epidemiología y Salud Pública (CIBERESP), 28029 Madrid, Spain; ivan.martinez.baz@navarra.es (I.M.-B.); mgcenoz@navarra.es (M.G.-C.); dtoledo@ub.edu (D.T.); pedro.plans@gencat.cat (P.P.-R.); carmen.munoza@sjd.es (C.M.-A.); jcastilc@navarra.es (J.C.); angela.dominguez@ub.edu (Á.D.); 3Instituto de Salud Pública de Navarra—IdiSNA, 31008 Pamplona, Spain; 4Agència de Salut Pública de Catalunya, 08005 Barcelona, Spain; iparron@gencat.cat (I.P.); joaquim.ferras@gencat.cat (J.F.); monica.carol@gencat.cat (M.C.); nuria.besm@gencat.cat (N.B.); nfollia@gencat.cat (N.F.); jpardosp@gmail.com (J.P.); isanz@mutuaterrassa.es (I.S.); mrosa.salaf@gencat.cat (M.-R.S.); jmendioroz@gencat.cat (J.M.); 5Agència de Salut Pública de Barcelona, 08023 Barcelona, Spain; mguillau@aspb.cat; 6Institut Català de la Salut (ICS), 08007 Lleida, Spain; 7Departament de Medicina, Universitat de Barcelona, 08036 Barcelona, Spain; 8Barcelona Tuberculosis Research Unit Foundation, 08008 Barcelona, Spain; joan.cayla@uitb.cat; 9Laboratorio de Microbiología, Hospital Sant Joan de Déu, 08950 Barcelona, Spain

**Keywords:** SARS-CoV-2, COVID-19, incidence rate, vaccine, household contact, secondary attack rate

## Abstract

The aim of this study was to evaluate the impact of index case vaccination on SARS-CoV-2 transmission to household contacts. In our epidemiological cohort study (May 2022–November 2023), we surveyed registered index case vaccination status and test results for contacts (testing on day 0, and on day 7 for negative contacts) and calculated the secondary attack rate (SAR), i.e., newly infected contacts/susceptible included contacts. The association of the independent variable, index case COVID-19 vaccination (yes/no), with household contact infection was determined using the adjusted odds ratio (aOR) and its 95% confidence interval (CI). We recorded 181 index cases and 314 contacts, of whom 250 agreed to participate; 16 contacts were excluded upon testing positive on day 0. Of the 234 included contacts, 49.1% were women, and the mean (SD) age was 51.9 (19.8) years. The overall SAR of 37.2% (87/234) was lower in the contacts of both vaccinated index cases (34.9% vs. 63.2%; *p* = 0.014) and index cases with a previous SARS-CoV-2 infection history (27.0% vs. 46.3%; *p* = 0.002). Index case vaccination showed a protective effect against infection for their household contacts (aOR = 0.21; 95% CI: 0.07, 0.67). The household SAR was high when the Omicron variant circulated. Vaccinated index cases were less likely to transmit SARS-CoV-2 to their contacts.

## 1. Introduction

An ongoing global priority is reducing SARS-CoV-2 infection transmission and preventing severe COVID-19 [1]. Some studies suggest that most new SARS-CoV-2 infections develop in the home [2,3]. While observational epidemiological studies and systematic reviews have been conducted on the household secondary attack rate (SAR) [2,4], the results are very heterogeneous, and our understanding of household SARS-CoV-2 transmission remains incomplete [5].

Vaccination effectiveness (VE), defined as the attack rate in vaccinated individuals compared to the attack rate in unvaccinated individuals, is expressed as a percentage re-duction in infections in vaccinated individuals. While a public health priority is understanding VE as key to reducing household transmission, calculating VE presents several problems [1]. SARS-CoV-2 transmission requires three inputs: (i) an infected index case capable of transmitting the virus, (ii) a contact of the index case susceptible to infection (a possible secondary case), and (iii) a transmission event between the index case and the secondary case. Empirically separating VE estimates for index and secondary cases is complex because a previous infection may affect both the infectivity of an index case and the susceptibility of a contact to infection [6]. One approach to obtaining the corresponding VE estimates involves collecting information on infected index cases linked to their exposed contacts and studying the vaccination status of both cases and contacts along with other relevant transmission factors [7].

In terms of preventing hospital admissions and COVID-19 deaths, VE has been widely confirmed to be high [8,9]. However, in terms of preventing clinical and subclinical SARS-CoV-2 infection, VE has been notably low and, furthermore, is progressively decreasing [10,11], indicating that vaccinated individuals can acquire and transmit SARS-CoV-2.

Regarding SARS-CoV-2 transmission in households, while immune protection can be induced by previous infection or vaccination [6,10,11], infection is favored by factors such as exposure intensity (e.g., a shared bedroom), nonuse of non-pharmacological measures (e.g., face masks), and smoking [1,7,12,13].

Several studies suggest that SARS-CoV-2 transmission increases with the emergence of new variants [2,4]. This increase in transmission was especially evident with the emergence of the Omicron variant, with the possibility of new infections confirmed for vaccinated people and even in people with a history of previous infection [2,4]. The role of vaccines in reducing infection is especially interesting in households because of the multiple opportunities for transmission and high SAR [12].

There is evidence that despite not preventing infection in some people, influenza and COVID-19 vaccines can reduce severity and death [9,11]. It is also important to study the role of vaccinated people as a source of infection, taking into account the vaccination status of their contacts at home [5], for two reasons: first, to obtain updated estimates of COVID-19 transmission in households in a uniform period of circulation of a particular variant (e.g., Omicron) and its subvariants; second, to understand the transmission capacity of vaccinated people in a household given the high numbers of vaccinated people and the recommendation to stay at home once infected.

The hypothesis in our study was that vaccines, while they may not prevent infection of index cases at home, may have an effect in reducing infection transmission to household contacts. Given this hypothesis, the vaccination of index cases would play a relevant role in reducing household transmission and would support the recommendation to vaccinate contacts, especially if they belong to risk groups.

The aims of this study were, during a period when the SARS-CoV-2 Omicron variant was circulating, to estimate (1) household contact SAR; (2) the reduction in index case infectivity; (3) the reduction in household contact susceptibility to infection given VE in both index cases and contacts.

## 2. Materials and Methods

### 2.1. Study Design

We carried out an epidemiological cohort study of SARS-CoV-2 transmission of the household contacts of index cases in Catalonia and Navarre (Spain), between May 2022 and November 2023 (when the Omicron variant was circulating). In the 8 participating primary care centers, SARS-CoV-2 cases were identified and selected at the beginning of each week using rapid antigen testing (RAT) and/or real-time polymerase chain reaction (RT-PCR) testing.

### 2.2. Participants

Household contacts associated with COVID-19 cases were recruited in 8 primary healthcare centers (1 in Navarre and 7 in Catalonia). Primary care centers were selected in each epidemiological surveillance unit, according to convenience criteria, by public health officials attached to the corresponding epidemiological unit.

Patient inclusion criteria were as follows: cases positive for SARS-CoV-2 and household contacts who agreed to participate in the study and provided their oral consent (index cases and contacts, respectively). Excluded were individuals with severe and uncorrectable cognitive and visual disorders, and individuals with hearing disabilities that hindered their ability to complete interviews.

Index cases were defined as confirmed cases of SARS-CoV-2 in the previous 10 days in participating centers who had at least one household contact who agreed to participate. Household contacts were defined as contacts with the index case for at least 2 h in the period running from 2 days before index case diagnosis to confirmation.

### 2.3. Questionnaire Design

As the first step in designing the epidemiological questionnaires, a comprehensive literature review was conducted by the coordination committee [4]. The questionnaires were structured taking into account COVID-19 recommendations of the World Health Organization, European Centres for Disease Prevention and Control, and the Spanish Ministry of Health. The research team, composed of professionals with epidemiological and public health research experience, held a series of preliminary meetings to develop the questionnaires, including the different sections, questions, and number of included elements. Discussions focused on question relevance, consistency, completeness, and clarity, and questionnaire length. The final questionnaire versions were obtained after an iterative process of several revisions of the earlier drafts.

The final questionnaires contained the following sections: social and demographic data, comorbidities and risk factors, epidemiological information, and knowledge of COVID-19 and preventive measures. The questionnaires also included data on previous SARS-CoV-2 infection and COVID-19 vaccination status; these data were validated through electronic health record linkage with regional vaccination registers and epidemiological surveillance unit databases.

### 2.4. Data Collection

The questionnaires were administered to the index cases and their household contacts. To detect cases of secondary infection, contacts were followed up for 7 days from confirmation of the index case infection. All contacts took a RAT on day 0, and those who tested negative underwent RT-PCR testing at the end of follow-up (day 7), regardless of whether or not they were symptomatic.

Data were collected for cases and contacts as follows: demographic variables (age and sex); date of onset of first symptoms; specific symptoms; diagnostic tests (RAT, RT-PCR); exposure time to the index case; relationship with the index case (cohabitation with a partner, other); shared bedroom with the index case; vaccination history and dates; history of SARS-CoV-2 infection and dates; risk factors (comorbidities and smoking); and control actions following index case diagnosis (face mask use, hand washing, hydroalcoholic solution use, distancing, ventilation, and isolation).

Study variable data were collected in an initial face-to-face interview and a subsequent telephone interview. Vaccination history and COVID-19 data were verified from the medical records. Participants (both index cases and household contacts) who had been vaccinated in the previous 21 days and 7 days were considered vaccinated with a first dose and second dose, respectively. Due to the small number of single-dose index cases and contacts, VE was studied on the basis of participants having received at least 1 dose.

### 2.5. Sampling and Sample Size

In each of the participating primary care centers, the first confirmed cases that met the inclusion criteria were initially selected every 15 days. Subsequently, due to a reduced incidence of new cases, this criterion was expanded to the selection of cases every week with no limitation on number. The sample was composed of 234 household contacts. This sample size, which allowed us to estimate the SAR of household contacts with a precision (e) of ±6% for a 95% confidence interval (CI), was calculated according to the following formulas: n = Zα^2^ × p × (1 − p)/e^2^ and e = √ Zα^2^ × p × (1 − p)/n.

### 2.6. Statistical Analysis

The SAR, expressed as a percentage, was calculated as the number of infected contacts 7 days after symptom onset in the index case (numerator) divided by the number of included contacts (denominator). Index cases and infected contacts on day 0 were excluded from both the numerator and denominator. The dependent variable was SARS-CoV-2 infection in contacts (yes/no), the independent variable was household contact exposure to a vaccinated index case (yes/no), and the contact covariables were as follows: a previous history of SARS-CoV-2 infection (yes/no); vaccination (yes/no); smoking (yes/no); cohabitation with a partner (yes/no); shared bedroom with the index case (yes/no); and face mask use at home following index case diagnosis (yes/no).

Using a logistic regression model, the adjusted odds ratio (aOR) and the corresponding 95% CI were calculated to determine the association between contact exposure to the vaccinated index case (yes/no) and contact infection (yes/no). The variables studied in the multivariate logistic regression model were selected using the backward method, for a cut-off point of *p* < 0.2. The variables for household contacts and their interaction evaluated in the model were: exposure to the vaccinated index case, age group (years), sex, previous COVID-19, contact vaccination ≥ 1 dose, smoker, cohabitation with a partner, shared bedroom, face mask use, and number of household contacts.

Index case VE (in reducing transmission) and household contact VE (in reducing susceptibility) were both calculated as VE = (1 − aOR) × 100 with the corresponding 95% CI.

To detect a possible confounding effect of previous SARS-CoV-2 infection on the vaccination of index cases and contacts, we repeated the statistical analysis using a secondary variable with 4 categories: (0) nonvaccinated and no previous infection; (1) vaccinated and no previous infection; (2) nonvaccinated and previous infection; and (3) vaccinated and previous infection (Appendix A). The aOR for each secondary variable category was calculated using the backward method and selecting the same variables as above (i.e., exposure to the vaccinated index case, age group (years), sex, previous COVID-19, contact vaccination ≥ 1 dose, smoker, and number of household contacts) for a cut-off point of *p* < 0.2 (Appendix A).

Analyses were performed using EpiInfo 7.2.5 (Centers for Disease Control and Prevention (CDC), Atlanta, GA, USA) and the SPSS v.24 statistical package (IBM, Armonk, New York, NY, USA).

### 2.7. Ethical Considerations

This study was approved by the Ethics Committee of the Arnau Vilanova University Hospital (code: CEIC-2464) and was conducted according to Declaration of Helsinki principles. All subjects included in the study received detailed information on the study aims and granted their consent to participate.

## 3. Results

Household contacts were studied for 181 index cases, with a mean (SD) age of 54.8 (19.1) years, 66.8% (121/181) of whom were women, and 97.8% (177/181) of whom had symptoms. In vaccination terms, 91.7% (166/181) had received at least one dose and 87.3% (158/181) at least two doses.

For the 181 index cases, 314 contacts were registered, of whom 250 agreed to participate. The exclusion of 16 contacts who tested positive to a RAT on day 0 left 234 contacts (Figure 1), with a mean (SD) age of 51.9 (19.8) years, 49.1% (115/234) of whom were women (Table 1). A high percentage—91.9% (215/234)—were contacts of vaccinated index cases, and, likewise, a high percentage had been vaccinated: 93.2% (218/234) with at least one dose and 87.2% (204/234) with at least two doses. Almost half—47.4% (111/234)—had a previous history of SARS-CoV-2 infection, 44.9% (105/234) were cohabiting with a partner, 38.5% (90/234) shared a bedroom with the index case, and 29.9% (70/234) were smokers. The overall SAR was 37.2% (87/234).

Regarding factors associated with SARS-CoV-2 transmission to household contacts (Table 2), no statistically significant differences were observed in the SAR for men compared to women (37.0% vs. 37.4%; *p* = 0.947). The SAR was higher in participants as follows: aged ≥65 years compared to aged ≤17 years (65.3% vs. 18.0%; *p* < 0.001); with no previous history of SARS-CoV-2 infection (46.3% vs. 27.0%; *p* = 0.002); smokers (51.4% vs. 31.1%; *p* = 0.003); cohabiting with a partner (49.5% vs. 27.1%; *p* = 0.001); and sharing a bedroom with the index case (44.4% vs. 32.6%; *p* = 0.069). No statistically significant differences were observed in the SAR for vaccinated and unvaccinated contacts (37.6% vs. 31.2%; *p* = 0.611).

In the multivariate logistic regression model (Table 3), the infection risk was lower in household contacts exposed to vaccinated index cases (aOR = 0.21; 95% CI: 0.07, 0.67) and with a previous SARS-CoV-2 infection history (aOR = 0.43; 95% CI: 0.23, 0.81) and was higher in contacts aged ≥65 years (aOR = 3.34; 95% CI: 1.00, 11.18) and in contacts cohabiting with a partner (aOR = 2.34; 95% CI: 1.08, 5.09). No protective role was found for household contact vaccination (aOR = 0.95; 95% CI: 0.23, 3.80). As for the possible confounding effect of previous SARS-CoV-2 infection on the vaccination of index cases and contacts, similar results, with only small differences, were obtained for the statistical analysis repeated using the secondary variable with four categories based on combinations of vaccinated, nonvaccinated, previous infection, and no previous infection (Appendix A).

## 4. Discussion

For the third year of the COVID-19 pandemic when the SARS-CoV-2 Omicron variant and its BA.4 and BA.5 subvariants were circulating, our main findings were that the household SAR was 37.2% and therefore, was among the highest transmission rates reported in meta-analyses [4,14], and that the SAR was lower in contacts exposed to vaccinated index cases. In terms of reducing the infection risk of household contacts, VE was 79% (95% CI: 33%, 93%) for vaccinated index cases but only 5% (95% CI: −280%, 67%) for vaccinated contacts. Other findings were that the SAR was much higher in participants aged ≥65 years and that a previous history of SARS-CoV-2 infection significantly reduced contact susceptibility.

The high SAR observed in our study is consistent with the increase in rates observed in the pandemic over time and with the emergence of new variants. López-Muñoz et al. [15], for their household contact study, estimated 58.2% SAR and 80.9% SAR for periods dominated by the Delta and Omicron variants, respectively. Madewell et al. [14], in an update of their systematic review of household SAR, estimated 42.7% SAR (95% CI: 35.4%, 50.4%) for periods dominated by the Omicron variant. While a direct comparison between variants is difficult, as vaccination levels and social restrictions varied, SAR estimates by Madewell et al. [14] for the Omicron (42.7%), Alpha (36.4%), and Delta (29.7%) variants were higher than the overall SAR of 18.9% previously reported for the earlier pandemic phase when the wild-type variant was prevalent. Transmission levels have been reported to be higher in households than in other community settings due to exposure intensity and multiple opportunities for transmission [9,16] and due to reduced use of protective measures such as face masks by vaccinated household contacts [15]. Most protocols still recommend home confinement for people infected with SARS-CoV-2 to reduce transmission in community settings, so studies of household transmission and control measures will continue to be a priority [16].

Various studies have indicated that vaccination, despite not preventing index case infection, may play a key role in reducing transmission to contacts [17] by reducing the viral load, symptoms, and even the number of transmission days in vaccinated individuals [18,19]. The high VE (79%) of index cases observed in our study points to a notable impact of vaccination in reducing household transmission, and underlines the especial importance of vaccinating people in contact with vulnerable populations, e.g., health workers, nursing home workers, individuals with frequent community contacts, and individuals cohabiting with elderly people and people at risk. Similar results have been observed in other studies that, using different methodologies, have estimated 40–80% reductions in household infection transmission [17,20,21].

In a U.K. study of secondary infection (defined as a positive SARS-CoV-2 test 2–14 days after a positive index case test), Harris et al. [5] compared risk for unvaccinated household contacts of infected persons who had been vaccinated at least once (ChAdOx1 nCoV-19 or BNT162b2) 21 days or more before testing positive with risk for unvaccinated household contacts of unvaccinated and infected persons; they reported that, overall, the transmission likelihood was around 40–50% lower in the households of vaccinated index cases and that results were similar for both vaccines. In a Dutch study, de Gier et al. [20] found that the household contact SAR was lower for fully vaccinated index cases than for unvaccinated index cases (11% vs. 31%), reporting an adjusted VE of 71% (95% CI: 63%, 77%). Eyre et al. [17], in their U.K. study, found that both the BNT162b2 and ChAdOx1 nCoV-19 vaccines were associated with reduced infection transmission from index cases that became infected despite vaccination (aOR = 0.32 and aOR = 0.48, respectively).

We found VE to be only 5% in terms of reducing infection susceptibility in contacts when the Omicron variant dominated; this corroborates the findings of other contact studies pointing to a comparative lack of VE for Omicron compared to the Delta and Alpha variants and pointing to a reduction in VE over time [21,22]. Some studies have reported superior vaccine protection in people with a history of previous SARS-CoV-2 infection (hybrid immunity). Suarez at al. [23] in France and Hall et al. [10] in the U.K. observed greater vaccine protection in previously infected individuals. In our study, greater vaccine protection was also observed in previously infected household contacts (aOR = 0.41; 95% CI: 0.07, 2.41), although this result was not statistically significant (Appendix A). Our finding is consistent with the results of a seroprevalence study conducted while the BA.4 and BA.5 Omicron subvariants were circulating: Castilla et al. [24] reported no change in COVID-19 risk in individuals who only had vaccine-induced antivirus spike (S) antibodies (suggesting that vaccines were ineffective in reducing infection susceptibility), whereas COVID-19 risk was significantly reduced in individuals with natural-infection-induced nucleocapsid (N) antibodies. The low VE observed for household contacts has also been associated with reduced use of nonpharmacological preventive measures by vaccinated individuals and higher exposure intensity and duration [16,25,26].

In our logistic regression model for both index case and contact vaccination, we found a statistically significant 57% effectiveness for previous infection; this rate is very similar to the 56% reported by Altarawaneh et al. [27] and the 51% reported by Suarez-Castillo et al. [23] and corroborates other findings of high protection due to infection-acquired immunity that could be increased with booster vaccination [6,10,28]. Regarding the effectiveness of booster doses, for a prospective cohort of healthcare workers, Hall et al. [10] found that, while infection-acquired immunity waned after 12 months in unvaccinated participants, it consistently remained above 90% in subsequently vaccinated participants, even if infected more than 18 months previously.

A result of our logistic regression model was that household contacts aged ≥65 years had a 3.3-fold greater risk of becoming infected compared to those aged ≤17 years, thereby corroborating the findings of various studies highlighting the greater infection risk of older household contacts [20,29,30], attributable to immunosenescence-related loss of protection from vaccines or previous infections [11,23,30,31].

Although our results were not statistically significant (probably due to the small number of subjects), we found that sharing a bedroom with an index case and smoking multiplied the infection risk 1.7-fold and 2.3-fold, respectively. Smoking, as has been confirmed for influenza, damages the respiratory immune system and increases general susceptibility to infection [13]. We found no evidence that face mask use by household contacts was effective in reducing infection. Note, however, that the role of face masks is difficult to establish because infection is transmitted before index case diagnosis, while contacts use face masks after index case diagnosis.

The main limitation of our study is that our sample was small and the statistical power to demonstrate VE in contacts was only 16%. Furthermore, the study’s capacity to demonstrate certain effects was constrained by the fact that over 90% of contacts had been vaccinated. The VE analysis was based on at least one dose, as most participants had received two or more doses (only under 5% of our index cases had received a single dose). While the index case vaccination effect on transmission could be confounded by a previous history SARS-CoV-2 infection, the effect was similar for index cases with and without this history (Appendix A).

Another limitation of our study is that data on exposure, risk factors (smoking), and use of non-pharmacological measures were not directly observed, so may reflect social desirability bias (i.e., respondents may have answered questions to be viewed favorably by the interviewer). Previous SARS-CoV-2 infections and vaccinations may also have been incorrectly reported, although this information was crosschecked against electronic health records (regional vaccination registers and epidemiological surveillance unit databases) and so can be considered as validated. Another issue is that potential contacts may have been rendered more susceptible to infection by less frequent use of non-pharmacological protective measures by vaccinated people due to the security instilled by vaccination, as reported elsewhere [15]. Although we collected previous SARS-CoV-2 infection data from medical records, previous infections may have been underestimated due to undiagnosed and unrecorded cases [24], while lower viral loads and less-severe clinical presentation of infections in vaccinated individuals would also have been more difficult to detect [19,32]. Furthermore, there is the risk that some infections that may have occurred outside the home were falsely attributed to the studied index cases, although this risk was minimized by household contacts being interviewed by contact tracing experts. Finally, people from the households who agreed to participate may not be representative of the overall set of households, and the number of participants may have been too small to uncover certain associations.

The strengths of this research are its prospective design of a household contact study of confirmed COVID-19 cases in a period of Omicron predominance and the fact that recruitment was based on a contact study protocol that remained unchanged during the study period and was applied to the entire population. Furthermore, participant variables were collected before knowing test results and contacts were classified according to their test results.

## 5. Conclusions

Our study shows that in the third year of the COVID-19 pandemic, dominated by the Omicron variant, the household SAR was high. The notable effectiveness of the index case vaccination in reducing household transmission points to the importance of prioritizing vaccination of groups in contact with at-risk populations and with frequent community contacts. According to our study, index case vaccination plays a key role in reducing household transmission, thereby supporting the recommendation of vaccination to protect contacts, especially if they belong to at-risk groups. Our study confirms that in the Omicron pandemic phase, household contact vaccination did not prevent infection [33]. More studies are needed to assess important household SAR factors, such as cohabitant vaccination and face mask use.

## Figures and Tables

**Figure 1 vaccines-12-00240-f001:**
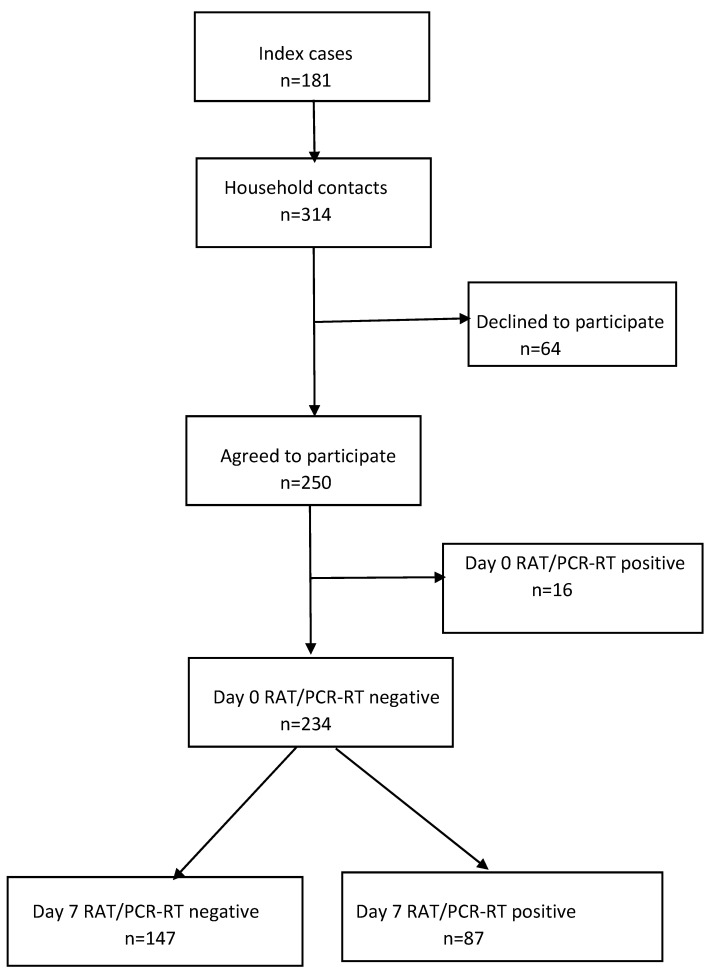
Study flow diagram. RAT, rapid antigen test; RT-PCR, real-time polymerase chain reaction.

**Table 1 vaccines-12-00240-t001:** SARS-CoV-2 secondary attack rate (SAR) in household contacts.

Variable	Infected Contacts n = 87	Total Contacts n = 234	SAR (%)
**Age group (years)**			
0–17	9	50	18.0
18–44	16	50	32.0
45–64	30	85	35.3
≥65	32	49	65.3
**Sex**			
Male	44	119	37.0
Female	43	115	37.4
**Index case vaccination**			
Yes	75	215	34.9
No	12	19	63.2
**Contact with previous COVID-19 history**			
Yes	30	111	27.0
No	57	123	46.3
**Smoker**			
Yes	36	70	51.4
No	51	164	31.1
**Contact vaccination ≥ 1 dose**			
Yes	82	218	37.6
No	5	16	31.2
**Cohabitation with partner**			
Yes	52	105	49.5
No	35	129	27.1
**Shared bedroom**			
Yes	40	90	44.4
No	47	144	32.6
**Face mask use**			
Yes	34	80	42.5
No	53	154	34.4
**Total**	**87**	**234**	**37.2**

**Table 2 vaccines-12-00240-t002:** Factors associated with SARS-CoV-2 transmission to household contacts.

Variable	Infected Contacts n = 74	Non-Infected Contacts n = 121	OR	95% CI	*p*-Value
Age ± SD	53.1 ± 21.0	39.0 ± 21.8	1.04		<0.001
**Age group (years)**					
0–17	9	41	1.00		
18–44	16	34	2.14	0.84–5.45	0.109
45–64	30	55	2.48	1.06–5.80	0.035
≥65	32	17	8.57	3.38–21.75	<0.001
**Sex**					
Male	44	75	0.98	0.58–1.67	0.947
Female	43	72	1.00		
**Index case vaccination**					
Yes	75	140	0.31	0.11–0.82	0.014
No	12	7	1.00		
**Previous COVID-19 history**					
Yes	30	81	0.43	0.25–0.74	0.002
No	57	66	1.00		
**Smoker**					
Yes	36	34	2.34	1.32–4.16	0.003
No	51	113	1.00		
**Contact vaccination ≥ 1 dose**					
Yes	82	136	1.32	0.44–3.95	0.611
No	5	11			
**Cohabitation with partner**					
Yes	52	53	2.63	1.52–4.54	<0.001
No	35	94	1.00		
**Shared bedroom**					
Yes	40	50	1.65	0.96–2.84	0.069
No	47	97	1.00		
**Face mask use**					
Yes	34	46	1.40	0.81–2.45	0.225
No	53	101	1.00		

CI: confidence interval; OR: odds ratio; SD: standard deviation.

**Table 3 vaccines-12-00240-t003:** Multivariate logistic regression of factors associated with SARS-CoV-2 transmission to household contacts.

Variable	aOR	95% CI	*p*-Value
**Age group (years)**			
0–17	1.00		
18–44	1.41	0.47–4.23	0.541
45–64	1.14	0.38–3.39	0.819
≥65	3.34	1.00–11.18	0.050
**Sex**			
Male	0.68	0.36–1.29	0.239
Female	1.00		
**Index case vaccination**			
Yes	0.21	0.07–0.67	0.008
No	1.00		
**Contact with previous COVID-19 history**			
Yes	0.43	0.23–0.81	0.009
No	1.00		
**Contact vaccination ≥ 1 dose**			
Yes	0.95	0.23–3.80	0.938
No	1.00		
**Smoker**			
Yes	2.32	1.18–4.54	0.014
No	1.00		
**Cohabitation with partner**			
Yes	2.34	1.08–5.09	0.031
No	1.00		

aOR: adjusted odds ratio (according to the remaining variables in the table and number of household contacts); CI: confidence interval.

## Data Availability

The data presented in this study are available on request from the corresponding author.

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
