# Peer review of "Vaccinated COVID-19 Index Cases Are Less Likely to Transmit SARS-CoV-2 to Their Household Contacts: A Cohort Study"

_vaccines, 2024, doi:10.3390/vaccines12030240_

Round 1

Reviewer 1 Report

Comments and Suggestions for Authors

Thank you for inviting me to review this manuscript. The manuscript reports a study entitled: Vaccinated COVID-19 index cases are less likely to transmit
SARS-CoV-2 to their household contacts. Although one should appreciate the authors' efforts, I am afraid to say the study lacks a clear question and theory. The most problematic issue with this submission relates to the methods section which needs extensive revisions. The section needs subtitles to clarify sampling and justification for sample size. In addition, there is a need for the vaccinated for COVID-19 cases which at present are described inadequately.  Finally I am not sure if the paper could appeal to international readers of the journal as the authors are reporting a very local topic. 

Comments on the Quality of English Language

The paper needs editing for the language.

Author Response

Response to Editor and Reviewers

Comments and Suggestions for Authors

Open Review -1

Thank you for inviting me to review this manuscript. The manuscript reports a study entitled: Vaccinated COVID-19 index cases are less likely to transmit SARS-CoV-2 to their household contacts.

Response. Thank you for reviewing the article and raising important questions. We have made further efforts to improve the manuscript.

Although one should appreciate the authors' efforts, I am afraid to say the study lacks a clear question and theory. The most problematic issue with this submission relates to the methods section which needs extensive revisions. The section needs subtitles to clarify sampling and justification for sample size. In addition, there is a need for the vaccinated for COVID-19 cases which at present are described inadequately. 

Response. In line with the reviewer’s recommendations, the entire Methods section has been reorganized into subsections that explain the questionnaires, the selection of primary care centers and participants, and the sampling system.

Finally I am not sure if the paper could appeal to international readers of the journal as the authors are reporting a very local topic.

Response. Thanks for the comment. However, note that the article refers to an area of southern Europe inhabited by around 8 million people, and we believe that our findings should be of interest to the journal's audience.

Comments on the Quality of English Language. The paper needs editing for the language.

Response. The English has been revised, as suggested.

Reviewer 2 Report

Comments and Suggestions for Authors

Dear Authors,

Thank you for submitting this paper that aims to estimate the likelihood of vaccinated index cases of SARS-CoV-2 transmitting the virus to their household contacts. The study also aimed to estimate COVID-19 vaccine effectiveness using the same study cohort. I can see the value of these types of studies, conducted in different populations/healthcare systems in helping to improve the evidence base as standalone findings or by contributing to future systematic reviews and metanalysis of SAR and VE estimates.

Title

I suggest that the study design be stated explicitly in the title. It will support bibliographic indexing and help the reader quickly determine the approach used.

Introduction

The authors state that the need for this study was triggered by the heterogeneous findings in previous studies conducted to estimate secondary attack rates (SAR) and the incomplete understanding of household transmission of SARS-CoV-2.
Based on this identified need, it is unclear to me how the study hypothesis and the aims of the study help to address the gap. This study as reported does not appear to undertake a much deeper exploration of transmission dynamics in the household setting but rather replicates the approach taken by previous studies that the authors referenced. It will be helpful if the authors can expand on this in the introduction to better highlight how this study aims to address this gap in our knowledge and understanding.

Materials and Methods

Page 2, lines 83-84: the three-pronged aims of the study requires that the sample size needed to achieve adequate power would have been considered a priori. The authors have not outlined how the study sample size was determined a priori. If this was not done a priori, was any calculation done after recruitment to determine the actual power of the study to examine the study hypothesis?

Page 2, line 87: please state what type of study design was used as the term "prospective epidemiological study" is vague. Is this an observational cohort study?

Page 2, lines 89-90: How were these primary care centers recruited? How many were originally invited and how many declined? Was there a systematic difference between those that accepted and those that declined that could impact on the type of study participants that were recruited?

Among the centers that agreed to participate, how did the authors select the eight centers? Were they selected randomly or opportunistically?

Page 2, lines 90-92: How were the index cases selected? Were they selected randomly?

When the authors state that index cases were selected using RAT and RT-PCR, do they mean new cases that were identified each week at these primary care centers were then invited to participate?

Page 2, lines 98-99: How were the two questionnaires developed and validated?

Page 3, lines 103-108: The authors helpfully list some of the data items collected using the questionnaires but it will also be helpful to know how some of these responses were independently verified to minimise mis-classification bias. For example, was the vaccination status & dates reported by the cases and contacts checked using existing clinical records held by the health centers?  In the discussion section, the authors allude to a similar check undertaken for reports of previous COVID-19 infection.

Page 3, lines 103 -108: Can the authors please state why they did not attempt to collect data on other plausible exposures to a confirmed COVID-19 case(s) outside the household during the study period? This is a potential confounding variable (like the previous history of COVID-19 infection).

Page 3, line 110: I am unclear what the difference is between "personal" and "telephone" interviews. Is the former a face to face interview?

Page 3, lines 125 -133: Can the authors please provide more detail on the statistical approach taken to develop the logistic regression model and the approach taken to build the MV model using the secondary independent variable.  This should include a description of how they addressed confounders and effect modifiers.

Can the authors also separately explain how they calculated the VE for index cases and the VE  for contacts. Is there a reason why a 95% CI was not provided for the VE estimate?

Results

Page 3, line 143: the flow of study participants may be better described using a flow diagram

Discussion

Page 6, line 180: the VE estimate for vaccinated cases is reported here for the first time and was not previously reported in the results section. The value of the VE estimate for vaccinated contacts is not shown anywhere in the paper and only reported as not being effective "in reducing the infection risk of household contacts". There is very little focus on the VE findings which is one of the three aims of the study.

Page 8, line 260: there are a few limitations of the study that I recommend the authors address directly in the discussion. It is not sufficiently clear how the authors tried to minimise or control for bias and confounding and this needs further explanation.

The authors allude to the small sample size of the study which questions how well powered this study is to examine the stated study hypothesis. It is also unclear how the sample size was determined and this raises concerns about selection bias.

Page 8, lines 278-280: the authors state that the strength of this study is its prospective design that was based on a contact study design. This was not mentioned in the Methods section and the contact study was not referenced in the discussion. If this is indeed a prospective cohort study, why did the authors not calculate risk and relative risk?

The discussion overall needs to be improved as it does not tell the reader how this study has succeeded in addressing the gaps in our knowledge articulated in the introduction. For example, the authors state that "Vaccination needs to be adapted to the circulation of new variants to reduce the susceptibility of household contacts" but the authors did not examine VE across time periods when different variants were dominant nor did they examine relative vaccine effectiveness (i.e. comparing VE of different vaccine types) in this study.

Best Wishes

Comments on the Quality of English Language

There are some grammatical errors that need to be corrected.

Author Response

Response to Editor and Reviewers

Comments and Suggestions for Authors

Thank you for submitting this paper that aims to estimate the likelihood of vaccinated index cases of SARS-CoV-2 transmitting the virus to their household contacts. The study also aimed to estimate COVID-19 vaccine effectiveness using the same study cohort. I can see the value of these types of studies, conducted in different populations/healthcare systems in helping to improve the evidence base as standalone findings or by contributing to future systematic reviews and metanalysis of SAR and VE estimates.

Response. Thank you for reviewing the article and raising important issues. We have made further efforts to improve the manuscript.

Title. I suggest that the study design be stated explicitly in the title. It will support bibliographic indexing and help the reader quickly determine the approach used.

Response. We have included information on the study design. The title is now “Vaccinated COVID-19 index cases are less likely to transmit SARS-CoV-2 to their household contacts: a cohort study”

Introduction. The authors state that the need for this study was triggered by the heterogeneous findings in previous studies conducted to estimate secondary attack rates (SAR) and the incomplete understanding of household transmission of SARS-CoV-2. Based on this identified need, it is unclear to me how the study hypothesis and the aims of the study help to address the gap. This study as reported does not appear to undertake a much deeper exploration of transmission dynamics in the household setting but rather replicates the approach taken by previous studies that the authors referenced. It will be helpful if the authors can expand on this in the introduction to better highlight how this study aims to address this gap in our knowledge and understanding.

Response. We have added the following at the end of the Introduction: “The role of people vaccinated against COVID-19 as a source of infection in households is relevant for two reasons. First to obtain updated estimates of Covid-19 transmission in households in a uniform period of circulation of the omicron variant and its subvariants; and secondly to know the transmission capacity of vaccinated people at home given that many people are vaccinated and it is recommended that they stay at home when they become infected.”

Materials and Methods. Page 2, lines 83-84: the three-pronged aims of the study requires that the sample size needed to achieve adequate power would have been considered a priori. The authors have not outlined how the study sample size was determined a priori. If this was not done a priori, was any calculation done after recruitment to determine the actual power of the study to examine the study hypothesis?

Response. We have included a new section on “Sampling and sample size” and have added “… The sample was made up of 234 household contacts, which allowed estimating the SAR in contact household with a precision ±6% and a 95% confidence interval (CI)”.

Page 2, line 87: please state what type of study design was used as the term "prospective epidemiological study" is vague. Is this an observational cohort study?

Response. We have clarified as follows: “We carried out an epidemiological cohort study of SARS-CoV-2 transmission in household contacts of index cases…”

Page 2, lines 89-90: How were these primary care centers recruited? How many were originally invited and how many declined? Was there a systematic difference between those that accepted and those that declined that could impact on the type of study participants that were recruited? Among the centers that agreed to participate, how did the authors select the eight centers? Were they selected randomly or opportunistically?

 Response. We have included a new section on “Participants” and have added “Recruitment of household contacts was carried out in eight primary health care centers (one in Navarra and seven from Catalonia) associated with COVID-19 cases. A primary care center was selected in each epidemiological surveillance unit according to convenience criteria by the public health officials of the corresponding epidemiological unit. The inclusion criteria are as follows: patients who are positive for COVID-19 (cases) and household contacts, who agree to participate in the study and provide oral consent. Those with the presence of severe and uncorrectable cognitive, visual or visual disorders. Hearing disabilities that hinder participants' ability to complete interviews are excluded from the study”.

Page 2, lines 90-92: How were the index cases selected? Were they selected randomly? When the authors state that index cases were selected using RAT and RT-PCR, do they mean new cases that were identified each week at these primary care centers were then invited to participate?

Response. In a new section on “Sampling and sample size” we have added “In each of the participant primary care centers, the first confirmed case that met the inclusion criteria was selected every 15 days. Subsequently, due to the reduction of incidence of new cases, this criterion was expanded to the selection of cases every week without limitation on number…”.

Page 2, lines 98-99: How were the two questionnaires developed and validated?

Response. A new section called “Questionnaire design” includes the following: “The first step in designing the questionnaires, a comprehensive literature review, was carried out by the coordination committee [4]. After this literature review, the questionnaires were structured, taking into account recommendations against COVID-19 provided by the World Health Organization, ECDC, and the Spanish Ministry of Health. The research team, made up of professionals with experience in epidemiological and public health research, held a series of preliminary meetings to develop all sections of the survey, including the selection of questions and the number of elements included. Discussion of the surveys focused on relevance, consistency, completeness, clarity of questions, and length of the surveys. The surveys were obtained after an iterative process that result-ed in several revisions of the first drafts.

The final surveys consisted of following sections: social and demographic information, comorbidities and risk factors, epidemiological information, COVID-19 knowledge about COVID-19 and its preventive measures. The questionnaires also included information on previous SARS-CoV-2 infection, and COVID-19 vaccination status, which could be validated through electronic health record linkage with the region-al vaccination registers and databases of epidemiological surveillance Units.”

Page 3, lines 103-108: The authors helpfully list some of the data items collected using the questionnaires but it will also be helpful to know how some of these responses were independently verified to minimise misclassification bias. For example, was the vaccination status & dates reported by the cases and contacts checked using existing clinical records held by the health centers?  In the discussion section, the authors allude to a similar check undertaken for reports of previous COVID-19 infection.

Response. As explained in “Questionnaire design”, “The questionnaires also include previous SARS-CoV-2 infection, and COVID-19 vaccination status, which could be validated through electronic health record linkage with the regional vaccination registers and databases of epidemiological surveillance”

Page 3, lines 103 -108: Can the authors please state why they did not attempt to collect data on other plausible exposures to a confirmed COVID-19 case(s) outside the household during the study period? This is a potential confounding variable (like the previous history of COVID-19 infection).

Response. Infections outside the home were only a hypothetical possibility encountered by the study interviewers. In the limitations we have explained: “ … some infections may have occurred outside the home and so may have been falsely attributed to the studied index cases. However, all household contacts were interviewed by professionals trained in contact tracing studies who ruled out community infection sources.

Page 3, line 110: I am unclear what the difference is between "personal" and "telephone" interviews. Is the former a face to face interview?

Response. We have clarified as follows: “Data on study variables were collected through a first face-to-face interview and completed with a telephone interview, and vaccination history and COVID-19 data were verified in medical records”.

Page 3, lines 125 -133: Can the authors please provide more detail on the statistical approach taken to develop the logistic regression model and the approach taken to build the MV model using the secondary independent variable.  This should include a description of how they addressed confounders and effect modifiers.

Response. This is explained in Methods: “The variables studied in the multivariate logistic regression model were selected using the backward method according to a cut-off point of p < 0.2. The variables of household contacts and their interaction that were evaluated in the model were: being exposed to vaccinated index case, age group (years), sex, previous history of COVID-19, vaccination contact ≥1 dose, smoker, living with couple, shared bedroom, use of mask and number of contacts in the household.”

“To detect a possible interference of previous SARS-CoV-2 infection in the effect of vaccination of index cases and contacts, we repeated the statistical analysis using a secondary variable with four categories: (0) Not vaccinated and without previous infection; (1) vaccinated and without previous infection; (2) Previous infection and not vaccinated; and (3) vaccinated and with previous infection (Supplementary tables 1 and 2). We calculated the aOR for each categories of this secondary variables using the backward method to select the same variables (being exposed to vaccinated index case, age group (years), sex, previous history of COVID-19, vaccination contact ≥1 dose, smoke) according to a cut-off point of p < 0.2.”

Can the authors also separately explain how they calculated the VE for index cases and the VE  for contacts. Is there a reason why a 95% CI was not provided for the VE estimate?

Response. This is explained in the Methods: “The index case VE to reduce transmission and the contact household VE to reduce susceptibility was calculated as VE=(1-aOR) x 100 with the corresponding 95% CI.”

Results. Page 3, line 143: the flow of study participants may be better described using a flow diagram

Response. A flow diagram of the study has been included as Figure 1.

 Discussion. Page 6, line 180: the VE estimate for vaccinated cases is reported here for the first time and was not previously reported in the results section. The value of the VE estimate for vaccinated contacts is not shown anywhere in the paper and only reported as not being effective "in reducing the infection risk of household contacts". There is very little focus on the VE findings which is one of the three aims of the study.

Response. Added in the first paragraph: “…the SAR was lower in contacts exposed to vaccinated index cases, which showed a VE of 79% (95% CI: 33% - 93%) in reducing the infection risk of household contacts; in contrast, contact vaccination showed no VE in reducing the infection risk of household contacts (5%; -280% - 67%)”.

Also, in the 4th paragraph of the Discussion we comment: “We found VE to be low (5%) in terms of reducing infection susceptibility in contacts in the period of Omicron variant dominance, corroborating other contact studies pointing to the lack of VE in preventing infection in vaccinated individuals when Omicron dominated compared to when the Delta and Alpha variants dominated, and also pointing to a reduction in VE over time [21],[22]”

Page 8, line 260: there are a few limitations of the study that I recommend the authors address directly in the discussion. It is not sufficiently clear how the authors tried to minimise or control for bias and confounding and this needs further explanation.

Response. We have added the following paragraphs in the limitations:

“The limitations of our study are that our sample was small and the statistical power to demonstrate the VE in contacts was only 16%.”

“Previous SARS-CoV-2 infections and vaccine doses could be incorrectly reported by contacts household but all questions related with previous SARS-CoV-2 infection, and COVID-19 vaccination status, would be validated through electronic health record linkage with the regional vaccination registers and databases of epidemiological surveillance units.”

“Furthermore, some infections may have occurred outside the home and so may have been falsely attributed to the studied index cases. However, all household contacts were interviewed by professionals trained in contact tracing studies and they ruled out community infection sources.”

The authors allude to the small sample size of the study which questions how well powered this study is to examine the stated study hypothesis. It is also unclear how the sample size was determined and this raises concerns about selection bias.

Response. Included in the Methods  is the following clarification: “The sample was made up of 234 household contacts, which allowed estimating the SAR in contact household with a precision ±6% and a 95% confidence interval (CI).” And also in the Discussion: “The limitations of our study are that our sample was small and the statistical power to demonstrate the VE in contacts was only 16%.”

Page 8, lines 278-280: the authors state that the strength of this study is its prospective design that was based on a contact study design. This was not mentioned in the Methods section and the contact study was not referenced in the discussion. If this is indeed a prospective cohort study, why did the authors not calculate risk and relative risk?

Response. As explained above, we have added in the Methods:  “We carried out an epidemiological cohort study of SARS-CoV-2 transmission in household contacts of index cases…”. Although it is a cohort study design, since we have calculated the risk of infection in the contact household as the accumulated incidence we think that use of the odds ratio is correct.

The discussion overall needs to be improved as it does not tell the reader how this study has succeeded in addressing the gaps in our knowledge articulated in the introduction. For example, the authors state that "Vaccination needs to be adapted to the circulation of new variants to reduce the susceptibility of household contacts" but the authors did not examine VE across time periods when different variants were dominant nor did they examine relative vaccine effectiveness (i.e. comparing VE of different vaccine types) in this study.

Response. In view of this comment, we have revised the Discussion and removed the quoted sentence.

Comments on the Quality of English Language. There are some grammatical errors that need to be corrected.

Response. The English has been revised, as suggested.

Reviewer 3 Report

Comments and Suggestions for Authors

This is an important study that looks at household transmission of COVID-19.

I enjoyed reading this manuscript. Here are my suggestions for improvement.

Abstract:

“The aim study was to evaluate the impact of index case vaccination on SARS-CoV-2 infection transmission to household contacts.”

Did you mean to say, “the aim of the study” and “vaccination status”?

“The SAR was lower in contacts of vaccinated index cases…”

Please define SAR in the abstract as well.

Also, please check grammar and spelling, as there are a lot of grammatical errors in the abstract.

Materials and Methods:

“An epidemiological questionnaire was administered to each index case and a specific questionnaire was administered to each contact “

Please specify what you mean by epidemiological questionnaire and specific questionnaire. What kind of questions were asked? What is the difference between these questionnaires?

“Using a logistic regression model and backward variable selection, adjusted odds ratio (aOR) and the corresponding 95% confidence interval (CI) values were calculated to 126

determine the association between contact exposure to the vaccinated index case (yes/no) and contact infection (yes/no).”

How did you control for the cluster effect when one index case had multiple contacts?

Also, it would be helpful if you mention software that was used for the data analysis.

Discussion:

Most of the discussion is about how this study results are consistent with previous studies. Perhaps it would be beneficial for the authors to emphasize a little bit more the unique aspects of this study, otherwise I get the impression that a sole purpose of this study was to validate what the previous studies had found on the topic.

Comments on the Quality of English Language

The quality of English could be improved, especially in abstract. 

Author Response

Response to Editor and Reviewers

Comments and Suggestions for Authors

Open Review -3

This is an important study that looks at household transmission of COVID-19. I enjoyed reading this manuscript. Here are my suggestions for improvement.

Response. Thank you for reviewing the article and raising important issues. We have made further efforts to improve the manuscript.

Abstract: “The aim study was to evaluate the impact of index case vaccination on SARS-CoV-2 infection transmission to household contacts.” Did you mean to say, “the aim of the study” and “vaccination status”?

Response. Thank you. We have reviewed the sentence, now rewritten as The objective of the study was to evaluate the impact of index case vaccination on SARS-CoV-2 transmission to their household contacts”

“The SAR was lower in contacts of vaccinated index cases…” Please define SAR in the abstract as well.

Response. Clarified as follows: “…the secondary attack rate (SAR) (new infected household contacts/susceptible registered household contacts)”

Also, please check grammar and spelling, as there are a lot of grammatical errors in the abstract.

Response. The English has been revised, as suggested.

 Materials and Methods: “An epidemiological questionnaire was administered to each index case and a specific questionnaire was administered to each contact “ Please specify what you mean by epidemiological questionnaire and specific questionnaire. What kind of questions were asked? What is the difference between these questionnaires?

Response. The index case and contacts questionnaires were the same except for the questions to the contacts to address exposure to the index case. We now explain this in the Methods:

“Questionnaire design

The first step in designing the questionnaires, a comprehensive literature review, was carried out by the coordination committee [4]. After this literature review, the questionnaires were structured, taking into account recommendations against COVID-19 provided by the World Health Organization, ECDC, and the Spanish Ministry of. The research team, made up of professionals with experience in epidemiological and public health research, held a series of preliminary meetings to develop all sections of the survey, including the selection of questions and the number of elements included. Discussion of the surveys focused on relevance, consistency, completeness, clarity of questions, and length of the surveys. The surveys were obtained after an iterative process that resulted in several revisions of the first drafts.

The final surveys consisted of following sections: social and demographic information, comorbidities and risk factors, epidemiological information, COVID-19 knowledge concerning COVID-19 and its preventive measures. The questionnaires also include previous SARS-CoV-2 infection, and COVID-19 vaccination status, which could be validated through electronic health record linkage with the regional vaccination registers and databases of epidemiological surveillance.”

“Using a logistic regression model and backward variable selection, adjusted odds ratio (aOR) and the corresponding 95% confidence interval (CI) values were calculated to determine the association between contact exposure to the vaccinated index case (yes/no) and contact infection (yes/no).” How did you control for the cluster effect when one index case had multiple contacts?

Response. We have explained in the Methods that the variable “number of household contacts of the index case” was used in the multivariate models to control for the cluster effect.

Also, it would be helpful if you mention software that was used for the data analysis.

Response. This information has been added: Analyses were performed using EpiInfo 7.2.5 and the SPSS v.24 statistical package.”

Discussion: Most of the discussion is about how this study results are consistent with previous studies. Perhaps it would be beneficial for the authors to emphasize a little bit more the unique aspects of this study, otherwise I get the impression that a sole purpose of this study was to validate what the previous studies had found on the topic.

Response. We have reviewed the conclusions as follows: Our study shows that, in the third year of the COVID-19 pandemic, dominated by the Omicron variant, the household SAR was high. The fact that index case vaccination was effective in reducing household transmission points to the importance of prioritizing vaccination of groups in contact with at risk populations and with frequent community contacts. In this period of Omicron variant, the vaccine of household contacts does not prevent infection”

Comments on the Quality of English Language. The quality of English could be improved, especially in abstract.

Response. The English has been revised, as suggested.

Round 2

Reviewer 1 Report

Comments and Suggestions for Authors

Thank you for inviting me to review this manuscript for the second time. Although the manuscript improved, I still wonder what is the message. Even the authors do not explain clearly what is Index case vaccination. The justification for sample size should be proved by using the sample size formula. I might be missing the importance of this study, thus remaining neutral to recommend rejection or acceptance.

Comments on the Quality of English Language

Acceptable.

Author Response

Open Review-1

Response to Editor and Reviewers

Comments and Suggestions for Authors

Thank you for inviting me to review this manuscript for the second time.

Response. Thank you for reviewing the article this second time and raising these important questions. We have made new efforts to improve the manuscript.

Although the manuscript improved, I still wonder what is the message.

Response. We have added the following text at the end of the Introduction: “Given this this hypothesis, the vaccine of index cases would have a relevant role in reducing transmission at home and would support the recommendation of vaccination to protect contacts, especially if they belong to risk groups.

In addition, we have added in Conclusion “According to our results, the vaccine of index cases would have a relevant role in reducing transmission at home and would support the recommendation of vaccination to protect contacts, especially if they belong to risk groups.

Even the authors do not explain clearly what is Index case vaccination.

Response. We have clarified in Methods as follows: “Participants, both index cases and household contacts, who had received the vaccination in the previous 21 days and 7 days were considered vaccinated with a first dose and a second dose, respectively. Due to the small number of single-dose index cases and contacts, VE was studied on the basis of participants having received at least 1 dose.”

The justification for sample size should be proved by using the sample size formula.

Response. This is explained in Methods as:  The sample was composed of 234 household contacts. According to the formulas: n = Zα2 x p x (1-p) / e2; e = √ Zα2 x p x (1-p) / n; this sample size allowed us to estimate the SAR of household contacts with a precision of ±6% for a 95% confidence interval (CI).

I might be missing the importance of this study, thus remaining neutral to recommend rejection or acceptance.

Response.Thank you

Comments on the Quality of English Language

Acceptable.

Response.Thank you

Reviewer 2 Report

Comments and Suggestions for Authors

Dear Authors,

Thanks for making the changes to the manuscript.

Author Response

Open Review-2

Response to Editor and Reviewers

Comments and Suggestions for Authors

Dear Authors,

Thanks for making the changes to the manuscript.

Response. Thank you for reviewing the article this second time